# Emotional skills and health assessment in interventions for intimate partner violence perpetrators: A systematic review of randomized controlled trials

Miguel Mora-Pelegrín[1], María Aranda[2]*, Beatriz Montes-Berges[2]

**1** Jaén Penitentiary Center, Jaén, Spain, **2** Department of Psychology/ Area of Social Psychology, University of Jaén, Jaén, Spain

* aranda@ujaen.es

## Abstract

The present study had two objectives: 1) To analyze the effectiveness of intervention programs for intimate partner violence (IPV) perpetrators that include learning emotional skills; and 2) to examine whether these programs assess variables related to the health and well-being of IPV perpetrators as a result of the intervention, and whether the obtained personal benefits are leveraged to enhance motivation towards therapy. This systematic review was carried out following the PRISMA guidelines. PsycINFO, PubMed, Scopus and PsycARTICLES databases were searched for articles published between January 2000 and December 2023, in order to identify randomized controlled trials of interventions for IPV perpetrators that included learning emotional skills, and that considered the health and well-being of the participants as motivational aspects in therapy. Fifteen primary studies that met the selection and eligibility criteria were included. The results indicated that slightly over half of the intervention programs for IPV perpetrators who develop emotional competencies were more effective in reducing recidivism and improving variables associated with this type of violence than were those that did not include this type of learning (53% of the trials showed significant differences in favor of the intervention groups). In addition, it became evident that the personal benefits obtained by the participants during therapy, which are related to their health and degree of well-being, are barely considered in efforts to improve motivational strategies. These results have important practical implications: firstly, they allow adjusting the contents of intervention programs for IPV perpetrators more effectively and, secondly, they add new elements that help the participants to improve their motivation and adherence to therapy.

**Data availability statement:** All relevant data are within the manuscript and its Supporting information files.

**Funding:** The author(s) received no specific funding for this work.

**Competing interests:** The authors have declared that no competing interests exist.

## Introduction

Intimate partner violence (IPV), the most common form of violence against women, continues to be a cause of concern worldwide. Estimates by the World Health Organization (WHO) indicate that almost one third of women aged 15–49 years report having suffered some form of physical and/or sexual violence perpetrated by their intimate partner [1]. Despite the high prevalence of this social phenomenon, offender intervention programs remain controversial for a variety of reasons related to their design, implementation, and outcome evaluation [2,3]. Several authors point out that therapeutic programs for abusers, which are among the main intervention strategies to prevent new acts of violence, mainly against women, are necessary to provide a comprehensive response to this problem [4–10]. Other authors maintain that a substantial percentage of the less serious episodes of IPV are perpetrated by women and that, therefore, this type of violence should be analyzed "in both directions" [11–14]. However, the effectiveness of the programs is unclear due to the lack of studies with control groups, artificially inflated improvements in short-term interventions, high dropout rates, and inconsistency of results when analyzing long follow-up periods [2]. Additionally, the limited evidence is mainly focused on assessing overall recidivism (despite contradictions when comparing official data with data from the victims themselves) and, to a lesser extent, on other variables related to improvements in some perpetrators' competences (e.g., emotion regulation, empathy, anger management, conflict resolution, etc.) [15–18]. In this sense, recent systematic reviews and meta-analyses reveal that the benefits of interventions for IPV perpetrators are not generalizable, since there are numerous limitations that should be taken into account; therefore, not all intervention proposals are robust and proven in terms of their effectiveness in reducing recidivism rates [2,19–21].

Previous review studies indicate that the intervention programs for IPV perpetrators that have offered the best results and largest effect sizes are those with a cognitive-behavioral approach [2,15,22–24]. This perspective considers violence as a learned behavior caused by cognitive distortions in relation to the partner, and aims to instill an alternative behavior to violence in abusers through the training of certain skills that lead to prosocial behavior and improved psychological well-being. Of particular interest are interventions that fall within the framework of the risk-need-responsivity (RNR) model, which promotes tailoring the intervention to consider the risk of recidivism, and the risk factors and responsiveness of offenders, as opposed to the more traditional "one-size-fits-all" intervention approach [25,26]. Other intervention programs are based on the Duluth Domestic Abuse Intervention Project or Duluth Model; influenced by feminist theory, which maintains that the primary cause of IPV is patriarchal ideology, these interventions are oriented towards the psycho-education of IPV perpetrators through consciousness-raising exercises to challenge the man's perceived right to control or dominate his partner, and a gender perspective that aims to encourage more egalitarian attitudes [10]. In practice, these approaches are mixed, and interventions often combine elements of both approaches [10,15,25], frequently including the learning of emotional skills (e.g., recognition and expression of emotions, emotional

self-regulation, empathy). However, such skills are not always prominent in therapy, despite their importance in achieving personal and social competences [10,27,28]. In line with previous reviews, although most intervention programs for IPV perpetrators include some strategies or activities aimed at fostering emotional skills, these are not typically a primary focus. As a result, there is generally no specific evaluation of these components, nor systematic follow-up on the effectiveness of emotional skills training in relation to outcomes such as recidivism [15–17]. On the other hand, motivational strategies (e.g., stage-based treatments, motivational interviewing, personalized feedback, the strengths-based approach, retention techniques), which are used extensively in change-resistant populations such as IPV perpetrators, have been shown to be effective in improving the therapeutic alliance, increasing treatment adherence and reducing the high dropout rates that characterize this type of intervention [29–33]. Taken together, the results of this research provide no clear evidence of the superiority of any one intervention approach over the others, although there are elements that work and could increase the effectiveness of programs targeting IPV perpetrators. It therefore seems necessary to continue the discussion on the effectiveness of the different intervention approaches. Moreover, a more in-depth analysis of programs content and the specific components that most significantly contribute to successful outcomes would substantially advance research in this field [34].

As noted by numerous authors, it is widely accepted that difficulties in recognizing and managing emotional stimuli, together with a lack of empathy towards the victim, are characteristics identified in most IPV perpetrators [18,34–37]. Studies with male IPV perpetrators show a relationship among deficiencies in skills related to emotional intelligence [38], the high levels of alexithymia they present with and the episodes of IPV they are involved in [39–47]. Likewise, empathy, as a key element of emotional intelligence, is considered a fundamental factor for the success of therapy, providing the IPV perpetrator with a more realistic view of the harm caused to the victim and a connection with their feelings that could have a deterrent effect with respect to repeating the aggressive behavior [47,48]. Emotional skills and the ability to empathize seem to act as moderators of aggressive behavior [16,49,50] and are involved in numerous functions related to prosocial behavior that would be expected to be altered in IPV perpetrators; therefore, the lack of these competences could interfere with the mechanisms that regulate their behavior and predispose them to this type of violence against a partner [44]. Taken together, these factors underline the need to explore new variables (e.g., a lack of emotion regulation) that affect the propensity towards violence and to enhance interventions aimed at improving these skills among IPV perpetrators [2,15,17,36]. At this juncture, it is important to note that a review of recent innovations in interventions aiming to reduce violence and aggression in adults (including IPV perpetrators) points to training in emotion recognition and regulation as being among the approaches associated with major improvements in outcomes [51].

On the other hand, some research confirms that, when analyzing the effectiveness of therapy, at least to date, the effect of the intervention on variables related to the health (physical and mental), quality of life and emotional state of IPV perpetrators is rarely taken into account, with the analysis being mainly focused on measuring primary outcomes related to the reduced rates of violence and recidivism [15,45,52]. However, this positive effect on personal and social spheres as participants' progress in therapy could be used as a motivational element, which is crucial in intervention programs for IPV perpetrators to reduce the high dropout rates, improve adherence to treatment and, ultimately, generate a genuine desire for change in the abuser [31,32].

Given the lack of clear evidence of the effectiveness of intervention programs for IPV perpetrators, as well as the importance of emotional skills as moderators of violent behavior, and the usefulness of "valuing" the personal benefits obtained when participating in therapy, an updated and rigorous review is needed to achieve the following objectives: 1) to analyze the effectiveness of intervention programs for IPV perpetrators that include the learning of emotional skills; and 2) to identify research that assesses variables related to the health and degree of well-being of IPV perpetrators as outcome variables, and then to verify whether this positive effect of emotional interventions is leveraged to enhance motivation for therapy. To obtain evidence on the effectiveness of this type of program, this review included only randomized controlled trials (RCTs). This approach aims to address some of the limitations identified in previous studies analyzing the effectiveness of interventions, particularly those related to the control of certain biases [53,54]. Specifically, randomized controlled

trials (RCTs) have been recognized as an optimal method for establishing the efficacy of interventions due to their design, which reduces bias and controls for confounding variables. Moreover, focusing exclusively on a single intervention modality—in this case, RCTs—allows for a critical examination of issues related to research design, treatment approaches, and outcome measurement.

## Method

This systematic review was conducted according to the Preferred Reporting Items for Systematic Reviews and Meta-Analyses (PRISMA 2020) recommendations [55]. The systematic review was not registered and the protocol was not published.

### Eligibility criteria

In accordance with the objectives of this research, the following inclusion criteria were applied to the reviewed studies: a) they were RCTs; b) the sample included adult men, women or both; c) the intervention incorporated emotional skills; d) they reported details of the intervention type, the content of the sessions and the duration of the intervention program; e) they provided recidivism rates or described outcomes about perpetrators' self-reported difficulties or deficits in variables associated with IPV (e.g., adaptive, emotional and empathic skills); and f) they indicated the participant follow-up period over which the main outcomes were measured. No restrictions were applied regarding the characteristics of the control group, the type of therapy or the language of the publication. Records were assessed for eligibility in two stages. Firstly, R1 and R2 independently assessed all titles and abstracts against the eligibility criteria. Secondly, full-text articles of potentially eligible records were independently assessed by R1 and R2, and disagreements were resolved and consensus was reached through discussion with a third researcher (R3).

### Search strategy

A systematic search of the PsycINFO, PubMed, Scopus and PsycARTICLES databases was conducted, covering the period from January 2000 to December 2023. The search strategy involved combining relevant terms related to RCTs, intervention programs for IPV perpetrators, the learning of emotional skills, and the health and well-being of abusers. The most productive keywords were: (Intimate Partner Violence Intervention) OR (Batterers Intervention Program) AND (efficacy OR Assessment OR evaluation) AND ("Emotional Skills" OR "Emotional Abilities" OR Empathy OR "Emotional Intelligence" OR "Motivational Feedback" OR "Health Status" OR "Well-being" OR Recidivism OR Attrition OR Dropout) AND ("Randomized Controlled Trial"). Searches by the first author's name, and of the reference lists of the selected studies in addition to the reference lists of other recent reviews, were also conducted. One study ultimately included in the review was identified in this way [56].

### Data extraction

R1 and R2 conducted data extraction independently based on the predefined data extraction form by following the Template for Intervention Description and Replication (TIDIeR) [57]. Any discrepancy was resolved through discussion and consensus was reached by consulting the third reviewer (R3).

### Assessment of methodological quality

Two authors (R1 and R2) independently evaluated the methodological quality of the included trials using the Cochrane Risk of Bias tool [58], which covers various sources of bias: 1) random sequence generation, 2) allocation concealment, 3) blinding of participants and personnel, 4) blinding of outcome assessment, 5) incomplete outcome data, 6) selective outcome reporting, and 7) other bias. Disagreements were resolved by consulting the third reviewer (R3).

## Results

### Study selection

Database searches resulted in 6,985 records (Fig 1). After removing duplicates, 6,559 studies remained for title and abstract screening, among which 158 were identified for full-text retrieval based on the aforementioned criteria. In total, 144 articles were excluded based on the fact that they were not RCTs ($n = 120$), the sample did not include IPV perpetrators ($n = 2$), the intervention did not include the learning of emotional skills ($n = 16$), or outcomes of interest were not assessed ($n = 6$). The remaining 14 trials [10,59–71] and an additional trial retrieved through the references of relevant articles [56] were included in the qualitative synthesis. Therefore, a total of 15 studies were included in the systematic review (see Table 1).

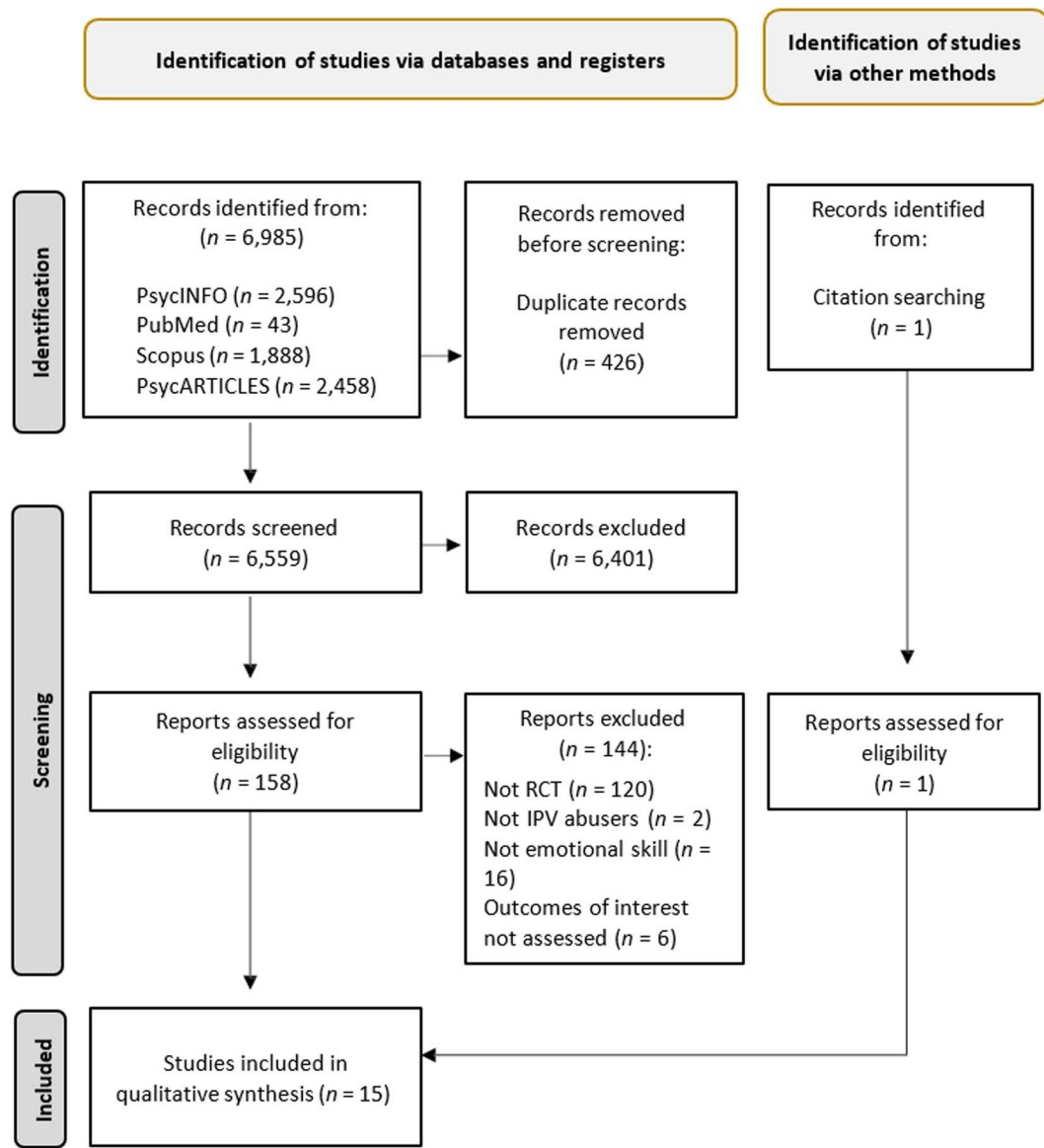

**Fig 1. Flowchart diagram.**

**Table 1. Main characteristics of the final sample.**

| | Sample size (N) Recruited population Mean age year (M) Court referred or Volunteers (% of men) | Intervention setting/ staff | Emotional intervention Intervention group (IG) | | | Standard or other intervention Control group (CG) | | Follow-up period | Outcomes (Main tools used) | Results |
|---|---|---|---|---|---|---|---|---|---|---|
| | | | Treatment type and intervention format (n) | Type of emotional skills delivered | N Sessions Intervention length | Treatment type and intervention format (n) | N Sessions Intervention length | | | |
| [59] Cavanaugh et al., (2011) USA | N = 55 attendees for a local anger management program (potentially at risk of IPV) M = 57.1% were between the ages of 18 and 35 years, and 42.9% were over 36 years old Volunteers (100% men) | Community centers/ Master's-level mental health professional | Dialectical Psycho-educational Workshop (DPEW) Group (n = 28) | Emotional regulation and Empathy | 1 session (2 hours) | Anger management program. Group (n = 27) | 1 session (2 hours) | 1 month (Evaluation feedback) | 1. Anger management skill (STAXI) 2. Empathy (BEES) 3. Self-controlling (WCQ) 4. Risk of eruptive violence (REV) | The IG increases awareness of adaptive coping skills, anger management, and empathy, and decreases potential risk for expressions of physical violence |
| [60] Cotti et al., (2020) USA | N = 154 convicted of intimate partner violence (misdemeanor offenders) M = 32.5 Court referred (60.4% men) | Community centers/ State-certified therapists | Cognitive Behavioral Therapy (CBT) Group (n = 77) | Emotions management (stress, anxiety and anger) | 16 sessions 9 months | Duluth model Group (n = 77) | 21 sessions 9 months | 3 years | 1. Official recidivism (OR) | IPV recidivism rate was 14 pp higher for offenders (men) assigned to CG |
| [61] Dunford et al., (2000) USA | N = 644 married U.S. Navy couples (misdemeanor offenders) M = 27 Court referred (Mostly men) | Military/ FAC- Navy staff and University of Colorado research team | Cognitive Behavioral Therapy (CBT) Group (n1 = 168) (n2 = 153) Individual (n3 = 173) | Emotions management (anger and jealousy) and Empathy | IG1–2: 32 sessions (26 weekly + 6 monthly) IG3: 12 monthly sessions 12 months | Individual security planning (n = 150) | 0 sessions 12 months | 6 months | 1. Official recidivism (OR) 2. Couples recidivism (CR) 3. Abusive behaviors (MCTS) | No significant differences between groups |
| [56] Easton et al., (2018) USA | N = 63 arrested for IPV with substance dependence (SUD) M = 39.4 Court referred (100% men) | Community centers/ Master's-level therapists | Substance Abuse-Domestic Violence therapy (SADV). Individual (n = 29) | Emotions management (anger and negative mood states) | 12 weekly sessions 3 months | Drug counseling Individual (n = 34) | 12 weekly sessions 3 months | 3 months | 1. State-Trait Anger Expression (STAXI) 2. Working Alliance (WAI) 3. Abusive behaviors (CTS-2) 4. Aggressive behaviors and hours of partner contact per day (TLFBSV) | SADV participants (IG) were less likely to engage in aggressive behavior proximal to a drinking episode, and reported fewer episodes of violence than CG participants at post-treatment follow-up. |

*(Continued)*

**Table 1.** (Continued)

| | Sample size (N) Recruited population Mean age year (M) Court referred or Volunteers (% of men) | Intervention setting/ staff | Emotional intervention Intervention group (IG) | | | Standard or other intervention Control group (CG) | | Follow-up period | Outcomes (Main tools used) | Results |
|---|---|---|---|---|---|---|---|---|---|---|
| | | | Treatment type and intervention format (n) | Type of emotional skills delivered | N Sessions Intervention length | Treatment type and intervention format (n) | N Sessions Intervention length | | | |
| [62] Fernández-Montalvo et al., (2019) Spain | N = 70 patients in a drug addiction program (SUD) with IPV episodes M = 35.7 Volunteers (58.6% men) | Community centers/ Clinical psychologists with 10 or more years of experience with addictions | Integrated intervention program for drug addiction and IPV. Group (n = 34) | Emotions management (anger and emotional expression) and Empathy | 36 sessions (with 20 additional 90-min IPV sessions) 12 months | Usual treatment in a drug addiction program (TAU). Group (n = 36) | 36 sessions (weekly-biweekly 45–60 min) 12 months | 6 months | 1. Couples recidivism (CR) 2. Abusive behaviors (CTS-2) 3. Distorted thoughts about the use of violence (IDT-V) 4. Anger management skill (STAXI) 5. Impulsivity (BIS-10) 6. Maladjustment Scale | The IG showed an IPV success rate significantly higher than CG and both groups achieved improvements in associated variables |
| [63] Gilchrist et al., (2021) UK | N = 104 offenders in substance use treatment M = 42.1 Volunteers (100% men) | Community centers (substance use service)/ Healthcare professionals | ADVANCE intervention (IPV + TAU) Individual/ Group (n = 54) | Emotional regulation and Empathy (perspective taking) | 16 sessions (2–4 individual sessions + 12 group sessions) 16 weeks | Usual substance use treatment (TAU) Individual/ Group (n = 50) | 16 sessions (Typically, fortnightly individual sessions + weekly group sessions) 16 weeks | Post-Intervention | 1. Couples recidivism (CR) 2. Intimate partner abuse (URICA-DV…) 3. Substance use (AUDIT…) 4. Mental health (PHQ-9, GAD-7…) 5. Self-control (BSC) | Neither substance use nor IPV perpetration had worsened for IG. Both groups improved in mental health, although the IG scored 10 pp less than the CG among those who scored highest on the scales |

*(Continued)*

| | Sample size (N) Recruited population Mean age year (M) Court referred or Volunteers (% of men) | Intervention setting/ staff | Emotional intervention Intervention group (IG) | | | Standard or other intervention Control group (CG) | | Follow-up period | Outcomes (Main tools used) | Results |
|---|---|---|---|---|---|---|---|---|---|---|
| | | | Treatment type and intervention format (n) | Type of emotional skills delivered | N Sessions Intervention length | Treatment type and intervention format (n) | N Sessions Intervention length | | | |
| [64] Hesser et al., (2017) Sweden | N = 65 participants who report significant difficulties with aggression (mild form) in a close stable relationship M = 36.9 Volunteers (43% men) | Internet/ Clinical psychology students with clinical training | Internet-delivered cognitive behavior therapy (iCBT) Individual (n = 32) | Emotion regulation (homework exercises) | Guided self-help format (therapists provided 24h feedback) 8 weeks | Monitored waitlist control Individual (n = 33) | 8 weeks | 1 year | 1. Violent behavior (CTS-2 and MMEA) 2. State/trail aggression (AQ) 3.Relationship quality (DAS) 4. Emotion regulation strategies (DERS) 5. Anger rumination (ARS) 6. Anxiety and depression symptoms (GAD-7 and PHQ-9) | The IG had significantly reduced emotional abuse, anxiety and depression symptoms. The treatment effect was partially mediated by changes in emotion-regulation ability |
| [65] Kraanen et al., (2013) Netherlands | N = 52 adults in SUD treatment with physical IPV M = 36.18 Volunteers (69% men) | Community centers/ Social workers with experience in substance abuse counseling | I-Stop (substance abuse and IPV) Individual (n = 27) | Emotion regulation (anger management and coping with emotions that may lead to substance use and IPV). | 16 sessions 16 weeks | CBT-SUD+ (only 1 IPV session) Individual (n = 25) | 16 sessions 16 weeks | Post-intervention | 1. Abusive behaviors (CTS-2) 2. Substance use (TLFP…) 3. Psychopathology (BSI) 4. Marital Satisfaction (MMQ) | No significant differences between groups (Both groups improve in substance use and IPV perpetration) |
| [66] Labriola et al., (2005) USA | N = 420 offenders arraigned on a domestic violence misdemeanor M = 31 Court referred (100% men) | Community centers/ Community centers staff | Duluth model (batterer program + monthly judicial monitoring; batterer program + graduated monitoring Group) (n1 = 102; n2 = 100) | Emotion regulation (recognizing emotions and controlling anger) | 26 weeks | Judicial monitoring (monthly and graduated) Individual (n1 = 109; n2 = 109) | 26 weeks | 1 year | 1. Official recidivism (OR) 2. Couples recidivism (CR) 3. Abusive behaviors (CTS) | No significant difference between groups |

*(Continued)*

| | Sample size (N) Recruited population Mean age year (M) Court referred or Volunteers (% of men) | Intervention setting/staff | Emotional intervention Intervention group (IG) | | | Standard or other intervention Control group (CG) | | Follow-up period | Outcomes (Main tools used) | Results |
|---|---|---|---|---|---|---|---|---|---|---|
| | | | Treatment type and intervention format (n) | Type of emotional skills delivered | N Sessions Intervention length | Treatment type and intervention format (n) | N Sessions Intervention length | | | |
| [67] Nesset et al., (2020) Norway | N = 125 offenders who voluntarily sought treatment M = 60.5 Volunteers (100% men) | Community centers (outpatient health service)/ Psychiatric nurses (for IG) and specialists in clinical psychology and education (for CG) | Cognitive Behavioral Therapy (CBT) Individual + group (n = 67) | Emotion regulation (dysfunctional anger and negative emotions) | 2 individual sessions + 15 group sessions 15 weeks | Mindfulness-based stress reduction Individual + group (n = 58) | 1 individual session + 8 group sessions 8 weeks | 1 year | 1. Couples recidivism (CR) 2. Abusive behavior (CTS-2) 3. Symptoms of emotional distress (HSCL-25) 4. General emotion regulation (DERS) | No significant difference between groups. A substantial risk estimate reduction was found in both groups. Both groups had a reduction of anxiety and depression (symptom scores remained high) and in difficulties in emotion regulation |
| [68] Romero-Martínez et al., (2022) Spain | N = 51 convicted of intimate partner violence M = 44.4 Court referred (100% men) | Community centers/ Therapists with one or more years of experience with batterer interventions | Standard Intervention Program (SIP) + Cognitive training Group (n = 20) | Empathic and emotion-decoding abilities | 35 (2h) + 31 (15 min) sessions (2xWeek) 9 months | SIP + Placebo training Group (n = 31) | 35 (2h) + 31 (15 min) sessions (2xWeek) 9 months | Post-intervention | 1. Processing speed (CPT-III) 2. Emotion decoding (Eyes test) 3. Risk of recidivism (SARA) | Only IG improved their processing speed and cognitive flexibility. The IG presented the lowest risk of recidivism. |
| [69] Rosenfeld et al., (2019) USA | N = 109 stalking offenders M = 36.05 Court referred (96% men) | Community centers/ Graduate students in clinical psychology trained to study | Dialectical Behavior Therapy (DBT – Adapted form of CBT to manage intense emotions) Individual + group (n = 57) | Emotional regulation (management of strong emotions and urges) | 48 sessions (24 group + 24 individual) 24 weeks | Cognitive Behavioral Therapy (CBT) Individual (n = 52) | 18 sessions 18 weeks | 6 months | 1. Official recidivism (OR) 2. Aggression (AQ) 3. Impulsivity (BIS-11) 4. Anger (STAXI) 5. Empathy (EQ) | No significant differences between groups on re-offense (recidivism). Small and significant effect in Empathy |

*(Continued)*

| | Sample size (N) Recruited population Mean age year (M) Court referred or Volunteers (% of men) | Intervention setting/ staff | Emotional intervention Intervention group (IG) | | | Standard or other intervention Control group (CG) | | Follow-up period | Outcomes (Main tools used) | Results |
|---|---|---|---|---|---|---|---|---|---|---|
| | | | Treatment type and intervention format (n) | Type of emotional skills delivered | N Sessions Intervention length | Treatment type and intervention format (n) | N Sessions Intervention length | | | |
| [70] Stover et al., (2019) USA | N = 62 fathers with a history of IPV who are under substance use disorder (SUD) treatment M = 35.85 Most of them were court referred (100% men) | Community centers/ Master´s-level clinicians specifically trained for the study | Fathers for Change (F4C). Integrated intervention IPV + child maltreatment + TAU Individual (n = 33) | Emotion regulation | 16 sessions (12 weekly sessions + 4 booster sessions) 12 weeks | Dads and Kids (DNK). Psycho-educational intervention (behavioral parenting skills). Individual (n = 29) | 16 sessions (12 weekly sessions + 4 booster sessions) 12 weeks | Post-intervention (3 months) | 1. Abusive behaviors (CTS-2 2. Difficulties in emotion regulation (DERS) 3. Anger management skill (STAXI) | No significant differences between groups on IPV. IG showed some benefits over CG in affect dysregulation symptoms and substance use relapse |
| [71] Zarling et al., (2015) USA | N = 101 adults in mental health treatment with IPV episodes M = 31 Volunteers (32% men) | Community centers (mental health services)/ Psychology doctoral students | Acceptance and Commitment Therapy (ACT). Group (n = 50) | Emotional Intelligence | 12 sessions 12 weeks | Support and discussion format. Group (n = 51) | 12 sessions 12 weeks | 6 months | 1. Emotional abuse (MMEA) 2. Physical aggression (CTS-2) 3. Experiential avoidance (AAQ) 4. Difficulties in emotion regulation (DERS) | The IG had significantly greater declines in psychological and physical aggression, and reductions in experiential avoidance and emotion dysregulation |
| [10] Zarling et al., (2022) USA | 338 convicted of IPV. M = 33.83 Court referred (100% men) | Community centers/ Qualified facilitators trained in ACT or Duluth Model approach | Acceptance and Commitment Therapy (Third-wave CBP to increase psychological flexibility) Group (n = 171) | Emotion regulation and Empathy (perspective-taking) | 24 sessions 24 weeks | Duluth model Group (n = 167) | 24 sessions 24 weeks | 1 year | 1. Official recidivism (OR) 2. Couples recidivism (CR) 3. Physical aggression (CTS-2) 4. Controlling behaviors (CBS) 5. Stalking behaviors (SBC) | No difference between groups in IPV charges (but victim reports indicated that IG engaged in fewer IPV behaviors). IG participants incurred fewer other charges. |

*Note.* ARS = Anger Rumination Scale; AQ = Aggression Questionnaire; AAQ = Avoidance and Action Questionnaire; AUDIT = Alcohol Use Disorders Identification Test; BEES = Balanced Emotional Empathy Scale; BIS-10 = Barratt Impulsiveness Scale; BSC = Brief Self Control Scale; BSI = Brief Symptom Inventory; CBS = Controlling Behaviors Scale; CG = control group; CPT-III = Continuous Performance Test-III; CTS-2 = Revised Conflicts Tactics Scale; DAS = Dyadic Adjustment Scale; DERS = Dysfunctional and Emotional Regulation Scale; EG = Empathy Questionnaire; GAD-7 = General Anxiety Disorder-7; HSCL-25 = Hopkins Symptom Checklist 25; IG = intervention group; IDT-V = Inventory of Distorted Thoughts About the Use of Violence;

*(Continued)*

**Table 1.** (Continued)

IPV = intimate partner violence; MCTS = Modified Conflicts Tactics Scale; MMEA = Multidimensional Measure of Emotional Abuse; MMQ = Maudsley Marital Questionnaire; PHQ-9 = Patient Health Questionnaire-9; REV = Risk or Eruptive Violence Scale; SARA = Spouse Assault Risk Assessment; SBC = Stalking Behavior Checklist; STAXI = State-Trait Anger Expression Inventory; SUD = substance use disorder; TAU = addiction treatment as usual; TLFB = Timeline Follow-Back Interview; TLFBSV = Timeline Follow-Back Spousal Violence; URICA-DV = Rhode Island Change Assessment for Domestic Violence Offenders-Revised; WAI = Working Alliance Inventory; WCQ = Ways of Coping Questionnaire.

## Study characteristics

The 15 papers selected for the systematic review included a total of 2,390 participants, with 1,369 in the intervention group and 1,021 in the control group. The sample sizes of these studies ranged from 28 to 644, and the mean age ranged from 27 to 60.5 years. All 15 studies included in the systematic review were published in English. Most of the trials were conducted in the USA (*n* = 9) [10,56,59–61,66,69–71], although one was conducted in the Netherlands [65], one in the UK [63], one in Norway [67], two in Spain [62,68], and one in Sweden [64]. Only two of the selected studies were published in the 2000s [61,66], while eight were published between 2010 and 2019 [56,59,62,64,65,69–71], and five from 2020 onwards [10,60,63,67,68].

## Quality and risk of bias assessment

A summary of the authors' judgements regarding the risk of bias for each included RCT is displayed in Fig 2. None of the trials satisfied all criteria. Thirteen of the fifteen trials (86.7%) met at least three criteria (20%) [65,66,70] or more (66.7%) [10,56,59,61–64,67,68,71]. Regarding random sequence generation, two trials (13.3%) did not clearly describe the method used to generate the randomization sequence in sufficient detail to allow assessing whether this produced comparable groups [60,69]. Knowledge of allocation concealment was clearly described, except in five trials (33.3%) that did not provide an adequate description [60,63,66,69,70]. Only seven trials (46.7%) reported blinding of participants and personnel [10,56,60,64,67,68,71], while the remaining eight trials (53.3%) provided unclear information about this domain or indicated that blinding was not possible [61–63,65,66,69,70]. Risk of bias was rated as low for five studies (33.3%) that ensured blinding of outcome assessors [56,60,63,64,67]. In the other 10 trials (66.7%), the available information was insufficient to assess the detection bias [10,59,61,62,65,66,68–71]. An intention-to-treat analysis was used in most trials (86.7%). Bias due to a lack of outcome data was unclear in only two studies (13.3%), where the missing outcome data (because of losses) may not have been similarly distributed between groups [60,69]. Concerning selective reporting, the information provided was insufficient to judge this domain in only four trials (26.7%), and the risk of bias was rated as medium [60,65,68,69]. Finally, the risk of other forms of bias was rated as medium or high in 11 trials (73.3%), since their baseline outcome measurements, baseline characteristics, or number of sessions were not similar; moreover, the training and involvement of the therapists could have been different, or protection against contamination may have been inadequate [10,56,60,62–67,69,70].

## Qualitative analysis

A total of 15 papers were included in accordance with the criteria outlined above. Table 1 summarizes the main characteristics of the studies: author name, publication year, country, sample characteristics (e.g., size, recruited population); intervention type (e.g., approach, format, number and duration of sessions, therapy providers); and outcome assessments (e.g., follow-up, results).

In line with the objectives of this study, some of the most relevant information from the reviewed research pertains to the types of interventions employed (including the underlying models, strategies, and skills targeted) and the outcomes reported. Data are also provided on the dropout rate, intervention doses, adherence to treatment, and effect sizes.

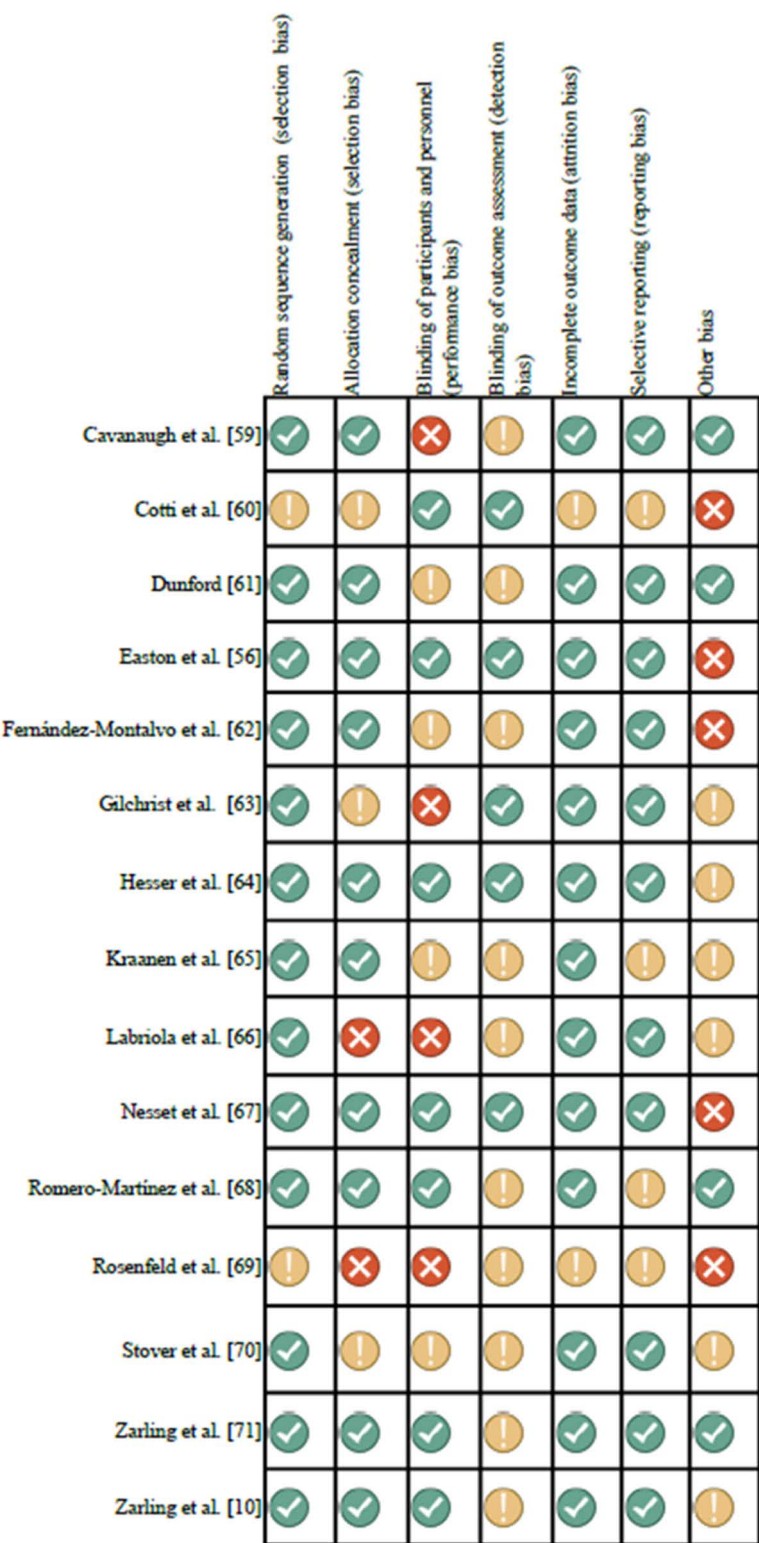

**Fig 2. Risk of bias for included trials.**

Regarding the intervention model, most trials (86.7%) based the intervention on cognitive behavioral therapy [10,56,59–62,64–71]. The Duluth Model was used for the intervention group in one trial (6.7%) [66] and, in another trial, the intervention was based on both approaches (6.7%) [63]. The content of the intervention specifically targeted IPV in 10 trials (66.7%) [10,59–61,64,66–69,71], while four interventions (26.7%) addressed both IPV and SUD [56,62,63,65], and one trial (6.7%) [70] integrated IPV, child maltreatment and SUD.

Training for a variety of emotional skills was provided in the RCTs. Emotion regulation was a component of all RCTs, most of which focused on the management of negative emotions and dysfunctional anger [56,60,62–67,69,70]. Some of the interventions targeted these factors along with emotional intelligence [71], pathological jealousy [61,62], negative mood states [56] or ruminative thoughts [64]. Six trials attempted to improve empathy [10,59,61–63,68]. Cavanaugh et al. [59] provided an abbreviated psychoeducational intervention focused on anger management, increasing the ability to feel and express empathy, and enhancing adaptive coping skills. Cotti et al. [60] carried out a program based on cognitive behavioral therapy, focused on improving intrahousehold behaviors and communication skills through counseling in relation to topics such as stress, anxiety, and anger management. Dunford [61] implemented three different 12-month interventions based on a cognitive-behavioral model, which included both didactic and process activities centered around empathy enhancement, communication skills, anger modification, and jealousy. Easton et al. [56] manually delivered individualized cognitive behavioral therapy for substance-dependent perpetrators of IPV, focusing on topics such as managing negative mood states, awareness of anger, management of anger related to significant others, and coping with criticism. Fernández-Montalvo et al. [62] proposed a manualized and integrated cognitive-behavioral treatment program for drug addiction and IPV that includes 20 additional sessions focused on aspects such as empathy, emotional illiteracy, anger management, anxiety/stress, depressive symptoms, pathological jealousy, and self-esteem deficits. Gilchrist et al. [63] delivered a 16-week integrated intervention for SUD and IPV over three temporal cycles, targeting improvements in emotional self-regulation, coping with stress, and perspective taking (empathy). Hesser et al. [64] delivered cognitive behavioral therapy via the Internet in the form of guided self-help involving a support therapist; the intervention incorporated emotion regulation and communication, anger rumination and conflict-resolution techniques. Mediation analysis revealed that the positive effects of treatment were partially mediated by changes in emotion regulation ability. Kraanen et al. [65] proposed an integrated treatment for SUD and IPV targeting anger management, communication skills, and coping with negative emotions and feelings. Labriola et al. [66] delivered an educational program over 26 weeks that included topics such as anger management, facilitating communication, and recognizing emotions. Nesset et al. [67] conducted cognitive-behavioral group therapy focused on dysfunctional anger, practicing communication, and training to accept and cope with negative emotions. Romero-Martínez et al. [68] implemented a 31-session cognitive training program, with brief exercises to improve the emotion-decoding abilities and empathic capacities of participants. Rosenfeld et al. [69] applied dialectical behavior therapy, tailored for stalking offenders, which emphasized learning skills to manage strong emotions and urges. Stover et al. [70] carried out an integrated intervention for fathers in residential SUD treatment targeting IPV and child maltreatment, and provided skills training to improve emotion regulation. Zarling et al. [71] implemented group-based acceptance and commitment therapy to reduce experiential avoidance and improve emotion recognition, expression, and regulation. Finally, Zarling et al. [10] implemented the same updated cognitive behavioral program to foster the learning of new ways to respond to emotions, among other abilities. However, despite the multiple strategies for working on emotional skills analyzed in the RCTs, there are few intervention programs in which the sessions aimed at promoting the acquisition of these skills represent a central aspect of the therapy or have special relevance. The emotional aspect clearly comprised the majority of the content of the treatment sessions in only four trials (26.7%) [10,59,64,71]. In the rest of the studies, the percentage of intervention sessions devoted to improving emotional skills was moderate (60%) [56,61–63,65–69] or low (13.3%) [60,70].

Regarding the main outcomes analyzed in the RCTs, behavior related to IPV was assessed in all trials. To assess physical and psychological IPV, the most widely used instrument was the Conflict Tactics Scales-revised (CTS-2) [72].

The Multidimensional Measure of Emotional Abuse (MMEA) [73] was included in two trials to measure emotional abuse [64,71]. Learning of emotional abilities was assessed in nine trials (60%) [56,59,62,64,67–71]. Four of these trials used the Difficulties in Emotion Regulation Scale (DERS) [74] to measure emotion dysregulation [64,67,70,71], although, for Nesset et al.'s [67] trial, these outcomes were reported in a separate publication [75]. Five trials used the State-Trait Anger Inventory (STAXI) [76] to measure anger management [56,59,62,69,70], and three trials included different tools, such as the Balanced Emotional Empathy Scale (BEES) [77] or the Eyes Test [78], to measure empathy and emotion-decoding abilities [59,68,69]. Finally, variables related to the health status of IPV perpetrators were assessed in only three trials (20%) [63,64,67]. The Patient Health Questionnaire-9 (PHQ-9) [79], the General Anxiety Disorder-7 (GAD-7) [80], and the Norwegian translation of the Hopkins Symptom Checklist 25 (HSCL-25) [81,82] were used in these studies to measure symptoms of emotional distress.

All trials analyzed recidivism rates or outcomes in relation to different factors associated with IPV. Of the eight trials (53.3%) that provided recidivism rates, two obtained official recidivism (OR) reports [60,69], three obtained couple or family reports [62,63,67], and another three provided both types of recidivism reports [61,66,71]. Seven of these trials also used self-report scales to measure variables associated with IPV [10,61–63,66,67,69]. The remaining seven trials (46.7%) assessed adaptive skills or the potential risk for IPV perpetration only via self-report scales [56,59,64,65,68,70,71]. The Spanish translation of the Spouse Assault Risk Assessment (SARA) [83,84] was used in one trial [68], and the Risk of Eruptive Violence Scale (REV) [85] was used in another [59].

Dropout and intervention doses were assessed in nearly all of the trials (86.7%). Only two trials (13.3%) [60,66] failed to report these aspects of the intervention or did not do so clearly. Adherence to treatment was high (> 50% of the participants completed at least 75% of the sessions) in 10 trials (66.7%) [10,56,59,61,62,64,67–69,71] and low (< 50% of the participants completed at least 75% of the sessions) in three trials (20%) [63,65,70]. Finally, the effect sizes were calculated in 10 trials (66.7%). Six trials reported large [59,71] or medium effect sizes [10,56,66,69], two trials reported effect sizes ranging from small to large [61,62], and two other trials reported effect sizes ranging from medium to large [65,68].

Regarding other characteristics of the 15 studies, the participant samples were diverse (e.g., male/female participants, misdemeanor/serious offenders, individuals with substance use disorders or mental health issues, court-referred/voluntary participants, etc.). Specifically, in terms of gender, over half of the trials (53.3%) recruited only male IPV perpetrators [10,56,59,63,66–68,70], while only two trials (13.3%) focused primarily on female IPV perpetrators [64,71]. Five trials (33.3%) included both male and female IPV perpetrators, with the majority of participants being men [60–62,65,69]. As for the intervention and staff setting, most trials (86.7%) implemented the interventions in community-based settings [10,56,59,60,562−71]. Only one trial was conducted in a military setting [61], and another was delivered online [64]. The interventions were facilitated by professionals from different fields, including healthcare providers [59], psychologists [62], therapists [10,60,66,68], and psychology students [64,69,71]. The therapists had specific experience in IPV interventions in only four trials (26.7%) [60,61,66,68].

The interventions were delivered on an individual basis in four trials (26.7%) [59,64,65,70], on a group basis in seven trials (46.7%) [10,59,60,62,67,68,71], and in both formats in four trials (26.7%) [61,63,66,69]. The duration and delivery format of the interventions varied across the reviewed studies (e.g., long-term, short-term, or single-session interventions; in-person or Internet-based treatments). Six trials (40%) implemented long-term programs consisting of more than 16 sessions [10,61,62,66,68,69], while nine trials (60%) delivered short-term interventions [56,59,60,63–65,67,70,71]. Following the implementation of the interventions, the reported follow-up periods varied considerably. The majority of trials (60%) included long-term follow-up assessments, ranging from 6 months [61,62,69,71] to 3 years [60]. The remaining trials (40%) reported short-term follow-ups, conducted either shortly after the intervention [59,63,65,68] or up to 3 months later [59,70].

Finally, all interventions included a control condition. In most studies, the control group received some form of intervention (e.g., standard SUD treatment [59,62,63,65], psychoeducation [10,60], CBT-based anger management [69], or

mindfulness-based approaches [67]). In two trials (13.3%), control group participants received a placebo training [68,71]. In three trials (20%), the control group received no treatment [61,64,66]. Most trials (53.3%) involved the same number of sessions in both groups [10,56,59,62,65,68,70,71], except for two trials (13.3%) that implemented fewer sessions in the intervention group [60,63], two other trials (13.3%) that implemented fewer sessions in the control group [61,63], and three trials (20%) where the control group received no treatment [61,64,66].

## Trial authors' findings and conclusions

Cavanaugh et al. [59] found a significant increase in awareness of adaptive coping skills, anger management, and empathy skills, and a decrease in the potential for the expression of physical violence, in the intervention group, as self-reported by participants at the 1-month follow-up. Cotti et al. [60] found that the official IPV recidivism rate was significantly (14%) higher for male offenders assigned to the control group, based on data from a 3-year natural field experiment. Dunford [61] revealed no significant differences between the experimental groups in a variety of outcome measures (OR, couple reports and self-reports) at the 6-month follow-up. Easton et al. [56] found that, based on self-reports, intervention group participants were significantly less likely to engage in aggressive behavior proximal to a drinking episode, and they reported fewer episodes of IPV than control group participants at the 3-month post-treatment follow-up. Fernández-Montalvo et al. [62] reported that their intervention group showed a significantly higher IPV success rate (the complete absence of physical, sexual, and/or psychological IPV episodes) than the control group, and both groups achieved statistically significant improvements in associated variables (based on self-reports and reports from participants' families) at the 6-month follow-up with respect to the baseline measurements. Gilchrist et al. [63] found that, at follow-up (16 weeks post-randomization), neither substance use nor IPV perpetration had worsened for men in the intervention group, and mental health improvements were also indicated by self-reports and the testimonies of their current or ex-female partners. Hesser et al. [64] described a significant reduction in emotional abuse, anxiety and depression symptoms (self-reports) in their intervention group at the 1-year follow-up. Mediation analysis using growth curve modeling revealed that the treatment effect was partially mediated by changes in emotion regulation ability. Kraanen et al. [65] found no significant differences in participant self-reports during an 8-week follow-up. Patients in both conditions significantly improved in terms of substance use and IPV perpetration. Labriola et al. [66] reported no significant differences between participants assigned to a batterer program or graduated monitoring in the probability of re-arrest for either offense at the 1-year follow-up. None of the interventions assessed resulted in a reduction in re-arrests or had a significant effect on victim self-reports. Nesset et al. [67] reported a substantial reduction in the estimated risk of violence in both groups at the 1-year follow-up, as indicated by self-reports and testimonies from female partners. In a different publication [75], a reduction in symptoms of anxiety and depression was reported, as well as a small but statistically significant reduction in difficulties in emotion self-regulation in both groups. Romero- Martínez et al. [68] found that only their intervention group improved in terms of processing speed and cognitive flexibility, and they also had a lower risk of recidivism after the intervention (self-report measurements). However, cognitive improvements and reductions in the risk of relapse after the intervention were unrelated. Rosenfeld et al. [69] reported no significant differences between groups in the re-offense rate (OR) at the 6-month follow-up, and a small but significant benefit of treatment on empathy self-reports. Stover et al. [70] found no significant differences between groups in IPV, although participants in the intervention group showed a significantly greater reduction in affect dysregulation problems and lower rates of substance use relapse (self-report measures) at the 3-month follow-up. Zarling et al. [71] described significantly greater declines in psychological and physical aggression among participants in the intervention group, and treatment-induced improvements were partially mediated by reductions in experiential avoidance and emotion dysregulation (self-report measures) at the 6-month follow-up. Finally, Zarling et al. [10] found no difference between groups in IPV charges at the 1-year follow-up, although participants in the intervention group acquired significantly fewer other violent and nonviolent charges, and their victims provided significantly lower scores on scales assessing IPV behaviors.

## Discussion

We conducted a systematic review of the effectiveness of emotional skills training in interventions for IPV perpetrators, considering the health and well-being of the latter as motivational aspects in therapy. A total of 15 trials were included in this systematic review to allow for a profound qualitative analysis of their results.

The interest in intervention programs for IPV perpetrators is reflected in the progressive increase in studies over recent decades, as noted in other reviews on this topic [3,51,86]. Specifically, one third of the trials selected in our study were published from 2020 onwards.

Overall, the results of this systematic review indicated that, for over half (53.3%) of the interventions for IPV perpetrators that included the acquisition of emotional skills, there was evidence of a reduction in recidivism and improvement variables associated with this type of violence compared with interventions that did not focus on developing these competencies (these trials showed significant differences in favor of the intervention groups) [56,59,60,62–64,68,71]. Furthermore, it became evident that the personal benefits obtained by participants in therapy, which are related to their health and state of well-being, are not usually assessed as outcome variables (only three of the analyzed trials [20%] did so [63,64,67]) and are scarcely taken into account as a means of improving motivational strategies in intervention programs (none of the trials did so). Regarding dropout rate reduction and intervention doses, the differences were not significant, although adherence to treatment was high in most of the trials analyzed. Therefore, we cannot confirm that interventions that focused on improving the emotional competencies of IPV perpetrators obtained better results in relation to this aspect of treatment.

Only one of the five trials that evaluated OR (all defined as new IPV charges) found evidence favoring emotional interventions for IPV perpetrators [60]. In two other trials, the lack of significant differences was justified by developing a rigorous and effective anger management program for the control group involving highly motivated therapists, which was considered superior to the usual intervention [69], or by the fact that the sample was considerably reduced due to the COVID-19 pandemic such that the ability to detect between-group differences was low for some of the criminal justice outcomes (.02 for IPV charges) despite significant improvements in the intervention group [10]. Regarding recidivism reported by partners or family members of IPV perpetrators, two trials reported success rates that indicate superiority of the emotional intervention over the standard or control intervention. In one trial, the success rate (total absence of IPV) reported by intervention group perpetrators and their family members was 60.7%, compared to 31.6% in the control group [62]. In the other trial, participants in the intervention group (perpetration) and their partners (victimization) scored higher on various scales used to measure abusive behaviors, controlling behaviors, communication patterns and attribution of responsibility for violence, among other factors [63]. In one of the trials that found no significant differences (estimated risk of violence reduced in both groups), the authors explained that this result could be due to the use of active therapy in the comparison condition, which shared an emotion regulation component with the intervention group [67]. In another trial, victim testimonies, contrary to what was suggested by the OR reports, indicated that the intervention group exhibited significantly less violent behavior towards victims than the control group [10]. It should be noted here that measuring recidivism is not without difficulty. OR, despite being the most objective and common way of measuring recidivism [19,87], is often defined heterogeneously and the criteria used to define recidivism are disparate [30]. Victim reports often point to higher rates of recidivism than information from official records or the self-reports of IPV perpetrators [2]. Nevertheless, as in one of the trials analyzed in this review, these reports may provide significant differences in favor of treatment interventions that are not reported by other sources [10]. In both cases, the rates may not be a true reflection of reality, as some offences may fall outside of criminal jurisdiction, be deferred, or never be reported by victims for a variety of reasons [2,19].

Regarding the data extracted from participants' responses, seven of the trials analyzed in this review (including two described above) [62,63] favored emotional interventions [56,59,64,68,71]. In several trials, no significant group differences were found (i.e., both groups improved), partly because, according to the authors, the effects of the intervention

may have been delayed and the post-treatment assessment did not allow firm conclusions to be drawn about the effectiveness of the compared treatments [65], or because participants in the control group had completed more treatment sessions. Moreover, some trials included an active comparison condition that made it difficult to observe differences between groups with small samples due to limited power, despite reporting a significant reduction in emotion-regulation problems in the intervention group [70].

In any case, in addition to the reasons cited above for the lack of significant group differences in this systematic review, it worth pointing out that it was difficult to discern which components of the treatment sessions completed by both groups were related to emotion recognition and management, due to the fact that either this was not clearly explained or the cognitive-behavioral and educational aspects could have been addressed by focusing on their emotional connection. This was evident in two of the trials, as negative emotion management was included in the treatment for the control group [67,69]. In general, attributing observed differences to the treatment received by the intervention group is difficult. In this systematic review, most of the control groups partook in an active (and heterogeneous, in many cases) treatment program compared to that completed by the intervention group. In three trials, the control group received no treatment at all, and in only five trials did control groups receive the usual treatment [62,63,65] or a placebo treatment to compare with the emotional intervention [68,71], which necessitates a cautious interpretation of the results.

Regarding the second main objective of our systematic review, the results showed that very few studies have investigated the effects of emotional intervention on the physical and mental health and well-being of IPV perpetrators. The results are in line with the review by Nesset et al. [67], which confirmed that self-assessment of the physical and mental health and quality of life of IPV perpetrators was barely addressed when investigating the effectiveness of therapy. In our study, only three trials assessed variables related to the health status of participants, and none of them leveraged the positive effect of the intervention to enhance motivation towards therapy. Gilchrist et al. [63] reported improvements in depression and anxiety in both their intervention and control groups at the end of treatment, although the intervention group scored 10% lower than the control group among those who scored highly on scales measuring both variables. Hesser et al. [64] described a significant reduction in anxiety and depression symptoms in their intervention group at the end of treatment, which was maintained at the 1-year follow-up (the control group received treatment after approximately 10 weeks post-treatment, thus the results from this group were not presented at the 1-year follow-up). Nesset et al. [75] reported a reduction in anxiety and depressive symptoms, and a small but statistically significant reduction in difficulties in emotional self-regulation, in both groups at the 1-year follow-up. As these authors point out, there is a paucity of research attempting to demonstrate that group-based interventions for IPV can improve the mental health of perpetrators and their partners. A recent meta-analysis indicated substantial associations of the various mental disorders studied with IPV perpetration and victimization [88], results that are supported by other meta-analyses that analyzed the association between mental health disorders and perpetration of IPV [89,90]. These results suggest that analyzing mental health problems in this context may lead to more effective individualized treatments [88]. In line with these ideas, we maintain that analyzing the health and well-being of IPV perpetrators will allow us to: a) determine the type of disorders that most frequently affect them, b) ascertain whether the appearance of these disorders coincided with the deterioration of their relationship, c) relate improvements in the management of emotions obtained via therapy with improvements in health and, finally, d) leverage the positive effect of therapy as motivation to improve adherence to treatment and reduce the high dropout rates that occur in intervention programs.

Finally, contrary to the results of other systematic reviews and meta-analyses [2,18], which reported a lack of efficacy of brief interventions, of the eight trials included in this study that reported significant improvements in participants in the emotional intervention groups, two of the intervention proposals were long-term [62,68], and the other six were short interventions [56,59,60,63,64,71]; this is more in line with the results reported by Babcock et al. [91], who found that short and long interventions were equally effective in terms of RO. Furthermore, in 50% of the trials that obtained significant

differences, these were maintained over longer periods of time ranging from 6 months to 3 years [60,62,64,71], suggesting that interventions that incorporate emotional content may produce longer-lasting benefits.

This systematic review is not without limitations. Although one of the strengths of our study is that we included only RCTs, we may have overlooked valuable findings from studies with other types of designs, such as observational studies or cohort studies. Furthermore, we acknowledge that including other types of studies could have enriched the discussion and offered a broader perspective. Another limitation pertains to statistical power; in some studies, the small number of participants that were ultimately analyzed made it difficult to calculate the effect size, as noted in Kraanen (2013) and Romero-Montalvo (2022). In any case, the small number of RCTs found in the literature reflects the difficulties in implementing this type of experimental method in interventions with IPV perpetrators [86,91–93]. In addition, four of the studies included in this review are pilot or feasibility trials for larger studies, and some of the trials did not clearly indicate whether the relevance of the emotional content in the interventions was high, which could limit the scope of the results. Moreover, the fact that almost half of the trials used only self-report measures means that the possibility of social desirability bias must be taken into consideration, where IPV perpetrators tend to minimize violent behaviors such that measures of intervention success may be distorted [19,93]. On the other hand, the inclusion of mixed samples of men and women in some trials, or of convicted men and voluntary participants, as well as the heterogeneity of various aspects of the methodology (e.g., format, duration, follow-up, specific training and experience of therapists working with IPV perpetrators), may make the interpretation of the results less certain. However, it is also true that not focusing on a specific type of IPV perpetrator (misdemeanor offenders and those potentially at risk of IPV were included), including samples with women (despite the fact that the most severe episodes of IPV are perpetrated mainly by men), as well including different methodological proposals could enhance the generalizability of the results. Future research should exhaustively define the contents of intervention proposals in order to better evaluate and compare the effects of treatment [18,19,21]. Furthermore, a meta-analysis of emotional intervention proposals aimed at IPV perpetrators could be carried out, partialling out the results of different samples and methodological approaches.

Despite its limitations, this systematic review provides a concise, rigorous and up-to- date synthesis of the effectiveness of intervention programs for IPV perpetrators that include the learning of emotional skills among their components, and it describes the difficulties and shortcomings in this area. In this sense, this review focused on novel aspects of the interventions, with the aim of highlighting the importance of empathy and the recognition and regulation of emotions as moderators of violent behavior against the partner. It also highlights the lack of studies assessing the health and well-being of IPV perpetrators when investigating the effectiveness of therapy, and the limited leveraging of the possible benefits of therapy to enhance motivation for treatment. Like other authors, we argue that innovative approaches need to be incorporated into IPV programs, which require radically different content from that which currently exists [17,54]. Moreover, interventions could be improved by adding emotional skills training [36,37].

Our final purpose was to identify ways to improve the effectiveness of intervention programs aimed at IPV perpetrators. For this reason, we aimed to relate the overcoming of emotional deficiencies as therapy progresses to the personal benefits (in health, well-being and the social environment) that can be obtained from such therapy. Lastly, through personalized and continuous feedback, the motivation of IPV perpetrators to adhere to treatment could be enhanced and, consequently, generate a genuine purpose for change.

## Supporting information

**S1 File. PRISMA Checklist.**
(PDF)

**S2 File. Supporting information.**
(ZIP)

## Author contributions

**Conceptualization:** Miguel Mora-Pelegrín, Maria Aranda, Beatriz Montes-Berges.

**Data curation:** Miguel Mora-Pelegrín.

**Formal analysis:** Miguel Mora-Pelegrín.

**Methodology:** Miguel Mora-Pelegrín, Maria Aranda.

**Supervision:** Beatriz Montes-Berges.

**Writing – original draft:** Miguel Mora-Pelegrín.

**Writing – review & editing:** Maria Aranda.

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
