## [Decision Letter · Decision Letter 0]

PONE-D-24-21114

Emotional Skills and Health Assessment in Interventions for Intimate Partner Violence Perpetrators: A Systematic Review of Randomized Controlled Trials

PLOS ONE

Dear Dr. Aranda,

Thank you for submitting your manuscript to PLOS ONE. After careful consideration, we feel that it has merit but does not fully meet PLOS ONE’s publication criteria as it currently stands. Therefore, we invite you to submit a revised version of the manuscript that addresses the points raised during the review process.

The article presents a comprehensive synthesis of the literature on an important topic: interventions for offenders. The search method is well-executed, and the results are thorough. However, as suggested by the reviewers, major revisions are needed before the paper can be considered for publication. Specifically, the theoretical background should be improved, and the objective of the literature review better defined. Additionally, the inclusion criteria need to be clarified.

We look forward to receiving your revised manuscript.

Kind regards,

Sara Ventura

Academic Editor

PLOS ONE

3. As required by our policy on Data Availability, please ensure your manuscript or supplementary information includes the following:

Additional Editor Comments:

The article presents a comprehensive synthesis of the literature on an important topic: interventions for offenders. The search method is well-executed, and the results are thorough. However, as suggested by the reviewers, major revisions are needed before the paper can be considered for publication. Specifically, the theoretical background should be improved, and the objective of the literature review better defined. Additionally, the inclusion criteria need to be clarified.

Reviewers' comments:

Reviewer's Responses to Questions

**Comments to the Author**

1. Is the manuscript technically sound, and do the data support the conclusions?

Reviewer #1: Yes

Reviewer #2: No

Reviewer #3: Yes

2. Has the statistical analysis been performed appropriately and rigorously? 

Reviewer #1: Yes

Reviewer #2: I Don't Know

Reviewer #3: N/A

3. Have the authors made all data underlying the findings in their manuscript fully available?

Reviewer #1: Yes

Reviewer #2: Yes

Reviewer #3: Yes

4. Is the manuscript presented in an intelligible fashion and written in standard English?

Reviewer #1: Yes

Reviewer #2: Yes

Reviewer #3: Yes

5. Review Comments to the Author

Reviewer #1: This is an interesting article that summarizes recent research regarding treatment interventions for male perpetrators of IPV. This article adds to the literature by examining outcomes for interventions that include learning emotional skills.

To be clear, is GBV used to mean intimate partner violence or any form of gender-based violence (including non-partner sexual violence)?

Introduction

• “which are among the main proposals to prevent new acts of violence against women”—"Proposals” does not quite seem to fit here—perhaps interventions, methods, options, or another synonym? (Proposals is also used on pg. 19.)

Results

• On page 13, programs ranging from 16- 56 sessions (weeks?) were considered long-term but a program with 12 sessions was considered short. Should those in a certain range (e.g., 20 or 23 weeks/ sessions and under) be considered “moderate” in length? (Length also mentioned on pg. 22.)

• Are most programs delivered once weekly for one hour? Do you have data regarding the dosage in terms of hours of treatment and number of sessions for the programs?

• The statement “one internet-based treatment (8 weeks) involving therapists who provided 24-hour feedback” feels like it could use more explanation. Was the program delivered live or self-paced and then the therapists provided feedback to the participants based on their submitted work? How does feedback work in IPV intervention programs?

• Regarding control groups, many of these groups were not recognized treatments for IPV. As such, one would not expect to see a change in IPV recidivism/ behaviour as the IPV is not being addressed. (“Concerning control groups, four interventions . . . focused predominantly on the usual SUD treatment . . . psychoeducational, focused on IPV . . . behavioral parenting skills plus SUD . . . anger management program . . . CBT anger management . . . mindfulness-based stress reduction therapy.” This seems that it would have different implications if the test groups included IPV tx + learning emotional skills versus control groups of IPV tx only. In this case, the test groups include IPV tx with emotional learning skills but the control groups do not appear to be IPV tx only.

• On page 18, it would be more clear to say, “Only one of the five trials that evaluated ORs found . . .”

Limitations

• “the inclusion of mixed samples of men and women in some trials”—it is not stated in the descriptive data how many studies also included women and the % of men in these samples. It should be stated above, in the information about the studies, how many studies included samples of 100% men, how many were mixed, and the %.

• “convicted men and voluntary participants” do you have data on how many studies included convicted, voluntary, or a mix?

• Good point: “not focusing on a specific type of perpetrator of GBV” Did any of the included studies mention perpetrator typology?

Additional resources

Regarding “programs that have offered the best results . . . cognitive-behavioral approach,” you may also find the following resources useful:

• Discussion of the RNR model in Bonta, J., & Andrews, D. A. (2024). The psychology of criminal conduct (7th ed.). Routledge. (or other publications by these authors)

• Radatz, D. L., & Wright, E. M. (2016). Integrating the principles of effective intervention into batterer intervention programming: The case for moving toward more evidence-based programming. Trauma, Violence, & Abuse, 17(1), 72-87. https://doi.org/10.1177/1524838014566695

• Stewart, L. A., Flight, J., & Slavin-Stewart, C. (2013). Applying effective corrections principles (RNR) to partner abuse interventions. Partner Abuse, 4(4), 494-534. http://dx.doi.org/ 10.1891/1946-6560.4.4.494

Thank you to the authors for their work on this topic!

Reviewer #2: Please see attached referee reports.

The choice of inclusion criteria (limiting attention to RCTs) limits this review from being able to speak to the body of work on batterer intervention programs, and the headline findings appear to be substantially overstating the results of the review.

Reviewer #3: This paper provides a systematic review of randomized controlled trials (RCTs) examining interventions for intimate partner violence (IPV) offenders. I have one primary concern and a suggestion for improving the study.

My main concern with this systematic review is that certain aspects of the research question are ambiguous, such as the study sample. Some studies included both male and female participants, others focused on individuals with diagnosed substance use disorders, some included court-referred participants, and one study involved participants from a military setting. Comparing intervention outcomes across such diverse populations is problematic, as it risks conflating fundamentally different participant groups—essentially comparing apples to oranges. The authors should provide more detail on how they account for these sample differences in review and in drawing their conclusions.

Moreover, to enhance the clarity of the review, it would be beneficial for the authors to include a summary table that outlines the key elements of each study (e.g., type of intervention program, main outcomes assessed, program duration, follow-up period, etc.), along with their findings and conclusions. Although these details are discussed in the text, a well-organized table would help structure the information and make it easier for readers to follow. Without such a table, it is challenging to track the findings of individual studies in relation to their other characteristics.

6. PLOS authors have the option to publish the peer review history of their article (what does this mean? ). If published, this will include your full peer review and any attached files.

**Do you want your identity to be public for this peer review?** For information about this choice, including consent withdrawal, please see our Privacy Policy .

Reviewer #1: No

Reviewer #2: No

Reviewer #3: No

---

## [Author Response · Author response to Decision Letter 1]

19 Dec 2024

Emotional Skills and Health Assessment in Interventions for Intimate Partner Violence Perpetrators: A Systematic Review of Randomized Controlled Trials

Manuscript ID PONE-D-24-21114

Dear Dr. Ventura,

Thank you for granting us the opportunity to further revise our manuscript titled “Emotional Skills and Health Assessment in Interventions for Intimate Partner Violence Perpetrators: A Systematic Review of Randomized Controlled Trials", and resubmit it to PLOS ONE.

We have carefully addressed the issues raised by the reviewers, and a detailed point-by-point response can be found in the uploaded file. Revisions to the manuscript have been marked in the upload file 'Revised Manuscript with Track Changes'. An unmarked version of our revised paper without tracked changes has been also uploaded in a separate file labeled 'Manuscript'.

If you have any additional questions or further requests, please do not hesitate to contact us. We look forward to your feedback.

Sincerely,

The authors

---

## [Decision Letter · Decision Letter 1]

PONE-D-24-21114R1Emotional Skills and Health Assessment in Interventions for Intimate Partner Violence Perpetrators: A Systematic Review of Randomized Controlled TrialsPLOS ONE

Dear Dr. Aranda,

Thank you for submitting your manuscript to PLOS ONE. After careful consideration, we feel that it has merit but does not fully meet PLOS ONE’s publication criteria as it currently stands. Therefore, we invite you to submit a revised version of the manuscript that addresses the points raised during the review process.

We look forward to receiving your revised manuscript.

Kind regards,

Sara Ventura

Academic Editor

PLOS ONE

Journal Requirements:

Additional Editor Comments:

This is a well-conducted and relevant systematic review that addresses a specific and underexplored aspect of the existing literature. I recommend minor revisions as specify by the reviewers, as the content is strong but a few structural and editorial adjustments are needed. Specifically: the title does not fully reflect the aims described in the abstract; terminology for IPV perpetrators should be used consistently throughout; the introduction would benefit from greater clarity regarding research gaps and the rationale for including only RCTs, along with additional references; in the methods section, certain inclusion criteria (e.g., theoretical approach) should be more clearly justified; in the results, redundancies should be avoided, and the detailed table currently in the supplementary materials should be integrated into the main text to improve readability; finally, all intervention types discussed in the review, including DBT, should be mentioned in the introduction. The discussion is well written and does not require substantial changes.

Comments from the editorial office: Upon internal evaluation of the reviews provided, we kindly request you to disregard the reviewer report provided by Reviewer 6. No amendments are required in response to reviewer 6’s comments

Reviewers' comments:

Reviewer's Responses to Questions

**Comments to the Author**

1. If the authors have adequately addressed your comments raised in a previous round of review and you feel that this manuscript is now acceptable for publication, you may indicate that here to bypass the “Comments to the Author” section, enter your conflict of interest statement in the “Confidential to Editor” section, and submit your "Accept" recommendation.

Reviewer #4: (No Response)

Reviewer #5: All comments have been addressed

Reviewer #6: All comments have been addressed

2. Is the manuscript technically sound, and do the data support the conclusions?

Reviewer #4: Yes

Reviewer #5: Yes

Reviewer #6: Yes

3. Has the statistical analysis been performed appropriately and rigorously? 

Reviewer #4: N/A

Reviewer #5: (No Response)

Reviewer #6: Yes

4. Have the authors made all data underlying the findings in their manuscript fully available?

Reviewer #4: Yes

Reviewer #5: Yes

Reviewer #6: Yes

5. Is the manuscript presented in an intelligible fashion and written in standard English?

Reviewer #4: No

Reviewer #5: Yes

Reviewer #6: Yes

6. Review Comments to the Author

Reviewer #4: Thank you for the opportunity to review ‘Emotional Skills and Health Assessment in Interventions for Intimate Partner Violence Perpetrators: A Systematic Review of Randomized Controlled Trials’. This is an important systematic review of the research literature describing RCT’s that address emotional skill building and/or measure health assessment of the IPV perpetrator. A cursory search of the published literature identified a few other systematic reviews and meta-analyses covering similar, but not this specific topic (Day, 2009; Eckhardt, 2013; Travers, 2021; Santirso, 2020; Karakurt, 2019), most of which are referred to in this manuscript. I conclude this was a worthwhile review contributing to a specific aspect of the literature than might impact future program effectiveness.

I am recommending that the manuscript be accepted with minor revisions as there are some structural and editorial issues, that would need to be addressed. The authors have a clear grasp on the subject matter and the discussion section makes good sense.

After reading and commenting on the paper, I then came across a revised version and I have done my best to incorporate feedback taking this into account. I had originally planned to say that the manuscript requires major revision of text for readability and flow, perhaps with assistance from a copy-editor, for the introduction in particular. For example, some of the arguments and statements seem to be lacking references. I also cannot understand why the main overview of studies is not presented within the study, as recommended by previous reviewers, but rather than as supplementary material. Specific comments are below.

1. Title. The title of the paper does not match the aim as described in the abstract (i.e. that the researchers were looking at health assessment also)

2. Terminology – I had comments about interchangeable use of terms for perpetrators also. My recommendation would be to further keep the definition to IPV perpetrator rather than switching terms (i.e. replace ‘aggressor’). I concur with earlier reviewer that ‘batterer’ is archaic and also not inclusive of the broad range of the population included in the review.

3. Introduction. A. The general structure of the introduction is okay, but some more specific explanation of the research gaps in terms of measuring effectiveness of emotional skill building (i.e. a sentence or two summarizing what has been done from reviews, etc., and the rationale for only including RCT’s would be useful.

B. The sentence commencing line 101 is problematic: ‘In this sense, it would be more productive to ask what content is truly effective and how it works, rather than focusing on a particular approach or type of intervention.’ I find hard to comprehend and implies that approach or intervention type are unimportant, whereas they could instead argue that further analysis of content would benefit intervention research. If indeed it is not so productive to discuss, why is there almost a page of text on this in the introduction?

C. Although revised down from the 100+ words in original version, sentence commencing line 64 still 50+ words long.

D. Sentence commencing line 129 lacking supporting references. I suggest checking throughout introduction for this issue.

1. Methods. A. Suggest changing ‘was not written down’ to ‘was not published’ – it is clearly written down within this manuscript!

B. The sentence commencing line 180 could be removed as it really belongs in results.

C. Line 156. States ‘d) they reported details of the theoretical approach’ – this is the first mention of this – if studies were only included if theoretical underpinning was described, I presume that many useful studies may have been excluded? If this is simply about terminology, then perhaps change this to intervention type or style to clarify. Otherwise, this idea needs to be included in introduction and the reason why theoretical approach was specified as an inclusion criteria clearly spelled out.

2. Results. A. Under ‘study selection’ the additional study identified via bibliographic review is mentioned both at the beginning and end of this section. One should be removed.

B. In general, there are large tracts of text that could instead be referred to in the table that is currently supplied as a supplementary material. I strongly encourage that the table is included within the manuscript, and that the description of the models, the study duration, the control/reference populations, and the follow-up periods could be reduced to 1-2 sentences each, with reference to the table. This would ensure the reader is more easily directed to the more pertinent results (i.e. the main outcomes and trial authors findings and conclusions)

C. Line 270 can you perhaps provide a different description of master-level therapist? I have no idea what this means – is it a counsellor or psychologist?

D. dialectical behavioral therapy was not presented in the introduction, and probably should as intro makes it looks like CBT and Duluth are the only approaches.

3. Discussion and conclusion. The discussion reads well and I had nothing to add to the comments and changes already made. The figures and table look fine too (I just think the table belongs in the body of manuscript).

Reviewer #5: Dear Author,

Thank you for your thorough revisions and thoughtful responses to all comments. The improvements made have significantly enhanced the clarity and contribution of the systematic review.

All previous concerns have been satisfactorily addressed. The additional explanations and refinements have strengthened the overall quality of the review.

At this stage, I have no further major concerns. Otherwise, I believe this paper is now ready for acceptance/publication.

Reviewer #6: High-Quality Bibliography:

The article demonstrates a remarkable effort in the selection of bibliographic references. The diversity of authors consulted indicates a strong command of the field of study and ensures a well-founded approach, both from the perspective of quantitative and qualitative studies, which is extremely important. The sources used are relevant, up-to-date, and reflect an ongoing dialogue with recent scientific production, while also incorporating classic contributions that shape the theoretical understanding of the topic. Without a doubt, this type of study provides an essential contribution to the issue under discussion.

Relevant Topic:

The central theme of the article is of undeniable social and academic relevance. It addresses an urgent issue that permeates various spheres of collective life—whether in the fields of mental health, justice, education, or human rights. The choice of topic not only justifies the significance of the study but also highlights the ethical and political responsibility of the researcher in addressing such a sensitive and contemporary problem. At a time when certain forms of violence and abuse remain normalized or rendered invisible, studies such as this are fundamental in breaking the silence and raising awareness.

Methodological Rigor:

The article presents a clear, well-justified, and properly executed methodological design. The methodological choices align with the research objectives, and the data collection and analysis procedures are described with precision. Moreover, there is evident concern in ensuring the validity of the results and the transparency of the entire investigative process.

Need for a Deeper Analysis of the Dynamics of Abuse:

One aspect that warrants greater attention is the depth of analysis regarding the dynamics of abuse itself. While the article presents important data and relevant reflections, the discussion could go beyond the mere exposition of facts, striving to uncover the subtle and complex mechanisms that sustain and perpetuate abuse in its various forms. A more critical analysis of the social and institutional contexts involved, as well as the power relations, silencing, and impunity that frequently surround such cases, would be desirable. Such deepening would provide a more comprehensive understanding of the phenomenon and enhance the study’s transformative potential.

Emphasizing the Social Impact of the Study:

Finally, it is crucial to highlight the need for a more explicit discussion of the study’s social impact. The article would benefit from a concluding section reflecting on the practical implications of its findings, suggesting avenues for social intervention, public policy formulation, or even institutional practice reform. By emphasizing the potential for change that studies like this can bring, the author strengthens the relevance of the work not only within academia but also in broader society. Scientific research gains even greater significance when it directly connects with real life and the possibility of constructing more just and safe realities.

Final Considerations:

In summary, this is a well-conducted study of great contemporary relevance that enriches scientific knowledge. Its theoretical, methodological, and social significance makes it an essential read for those dedicated to understanding and addressing contemporary forms of violence and abuse.

7. PLOS authors have the option to publish the peer review history of their article (what does this mean? ). If published, this will include your full peer review and any attached files.

**Do you want your identity to be public for this peer review?** For information about this choice, including consent withdrawal, please see our Privacy Policy .

Reviewer #4: No

Reviewer #5: No

Reviewer #6: No

---

## [Author Response · Author response to Decision Letter 2]

5 Jun 2025

Journal requirements’ and comments of the Editor

Thank you very much for the opportunity to revise our manuscript and for the valuable feedback provided by the reviewers. We greatly appreciate the thoughtful comments and constructive suggestions, which have contributed to improving the quality and clarity of our work.

We have carefully reviewed the manuscript and implemented the recommended revisions. Specifically, we have made changes related to the reference list, the abstract, and the terminology concerning IPV perpetrators. Revisions have also been made to the Introduction, Method, and Results sections to enhance the overall rigor and coherence of the manuscript.

In addition, we have prepared a detailed point-by-point response addressing each of the reviewers’ comments. Extended responses to specific concerns—particularly those raised by Reviewer 4—are included in the accompanying response document.

We hope that the revised version of the manuscript meets the expectations of the editorial team and reviewers, and we remain at your disposal for any further clarifications or modifications.

Reviewers’ comments to the authors and responses

Reviewer 4

Comment 1. The manuscript requires a revision of text for readability and flow, perhaps with assistance from a copy-editor, for the introduction in particular.

Response: We appreciate your observation regarding the readability and flow of the manuscript, particularly in the Introduction section. To ensure the quality of these revisions, we sought the assistance of a professional academic copy-editor, who provided detailed editorial support. The editor's certificate has been included as part of the manuscript submission.

Comment 2. […] the main overview of studies is not presented within the study, as recommended by previous reviewers, but rather than as supplementary material.

Response: Thank you for your comment. The main overview of the studies has been incorporated into the main text. A table titled “Summary of the Main Characteristics of the Final Sample” has been included in the Results section.

Comment 3 (Title). The title of the paper does not match the aim as described in the abstract (i.e. that the researchers were looking at health assessment also).

Response: Thank you for this insightful comment. It prompted us to reflect on how we had originally articulated the objectives in the abstract. As the reviewer rightly noted, the abstract did not adequately convey the equal importance of both objectives. This may have given the impression that examining whether programs use health improvement measures as indicators of change and as motivation for intervention adherence was a secondary aim. We have revised the wording to clearly state both objectives of the review. We hope this modification ensures greater consistency with the title and the content developed throughout the manuscript.

Comment 4 (Terminology). I had comments about interchangeable use of terms for perpetrators also. My recommendation would be to further keep the definition to IPV perpetrator rather than switching terms (i.e. replace ‘aggressor’). I concur with earlier reviewer that ‘batterer’ is archaic and also not inclusive of the broad range of the population included in the review.

Response: In accordance with the reviewer’s comment, we have reviewed the terminology used throughout the text and replaced the relevant terms with IPV perpetrators. To improve readability and avoid excessive repetition of the term within the same paragraph, we have also used the word participants where appropriate. Additionally, we consulted with the academic copy-editor for guidance on this matter.

Comment 5 (Introduction-A). The general structure of the introduction is okay, but some more specific explanation of the research gaps in terms of measuring effectiveness of emotional skill building (i.e. a sentence or two summarizing what has been done from reviews, etc., and the rationale for only including RCT’s would be useful.

Response: We appreciate your comment. Both aspects have been revised in the Introduction. We have specified and expanded upon the limitations of intervention programs with regard to the incorporation and evaluation of emotional skills. Concerning the use of RCTs, we have sought to strengthen the justification for their selection. In both cases, citations have been included to support the revisions.

Comment 6 (Introduction-B). The sentence commencing line 101 is problematic: ‘In this sense, it would be more productive to ask what content is truly effective and how it works, rather than focusing on a particular approach or type of intervention.’ I find hard to comprehend and implies that approach or intervention type are unimportant, whereas they could instead argue that further analysis of content would benefit intervention research. If indeed it is not so productive to discuss, why is there almost a page of text on this in the introduction?

Response: Thank you for your comment. Your observation is very timely and, indeed, we agree that clarifying this idea and introducing it from the perspective you suggest provides greater coherence with the manuscript’s line of argument.

Comment 7 (Introduction-C). C. Although revised down from the 100+ words in original version, sentence commencing line 64 still 50+ words long.

Response: In accordance with the reviewer’s comment, the sentence has been reduced.

Comment 8 (Introduction-D). Sentence commencing line 129 lacking supporting references. I suggest checking throughout introduction for this issue.

Response: Thank you for the comment. Indeed, the statement highlighted by the reviewer requires evidence-based support. To strengthen the argument, we have added two references: Santirso et al. (2020) [30], who discuss motivational strategies that enhance treatment adherence and emphasize the importance of “helping individuals find their own reasons for change”; and Soleymani et al. (2018) [31], who report that motivational interviewing improves motivation or readiness for change, engagement in treatment, and session attendance.

Comment 9 (Methods-A). Suggest changing ‘was not written down’ to ‘was not published’ – it is clearly written down within this manuscript!

Response: Changed in the text, as suggested by the reviewer.

Comment 10 (Methods-B). The sentence commencing line 180 could be removed as it really belongs in results.

Response: The sentence has been removed from the Methods section and incorporated—with adjusted wording—into the Results section.

Comment 11 (Methods-C). Line 156. States ‘d) they reported details of the theoretical approach’ – this is the first mention of this – if studies were only included if theoretical underpinning was described, I presume that many useful studies may have been excluded? If this is simply about terminology, then perhaps change this to intervention type or style to clarify. Otherwise, this idea needs to be included in introduction and the reason why theoretical approach was specified as an inclusion criteria clearly spelled out.

Response: We appreciate your insightful comment. We have replaced the term in the text with "intervention type." Indeed, that is the concept we intended to convey.

Comment 12 (Results-A). Under ‘study selection’ the additional study identified via bibliographic review is mentioned both at the beginning and end of this section. One should be removed.

Response: The duplicated information has been removed.

Comment 13 (Results-B). In general, there are large tracts of text that could instead be referred to in the table that is currently supplied as a supplementary material. I strongly encourage that the table is included within the manuscript, and that the description of the models, the study duration, the control/reference populations, and the follow-up periods could be reduced to 1-2 sentences each, with reference to the table. This would ensure the reader is more easily directed to the more pertinent results (i.e. the main outcomes and trial authors findings and conclusions)

Response: Thank you for your helpful comment. In response, we have integrated a summary table outlining the main characteristics of the studies directly into the text. Furthermore, following the reviewer’s suggestion, we revised the Results section to eliminate redundancy, with the aim of enhancing clarity and improving the manuscript’s overall readability. Specifically, priority has been given to information that helps address the objectives of our research and will later inform the discussion—namely, details related to the content of the interventions and their outcomes. The remaining information has been briefly summarized, with only selected aspects of interest presented at the end of that section.

Comment 14 (Results-C). Line 270 can you perhaps provide a different description of master-level therapist? I have no idea what this means – is it a counsellor or psychologist?

Response: Thank you for your comment. As the information in the Results section was reduced, the concept “master-level therapist” has been removed.

Comment 15 (Results-D). Dialectical behavioral therapy was not presented in the introduction, and probably should as intro makes it looks like CBT and Duluth are the only approaches.

Response: Thank you very much for your comment. Dialectical Behavior Therapy (DBT) is a form of cognitive-behavioral therapy (CBT) that emphasizes the development of concrete skills, particularly those aimed at learning to manage intense emotions and impulses. DBT was not specifically discussed in the introduction because its core components and foundational principles are derived from CBT, which was addressed in that section. To clarify this point, Table 1 includes this information so that readers can properly situate DBT within the broader framework of CBT.

Comment 16 (Discussion and conclusion). The discussion reads well and I had nothing to add to the comments and changes already made. The figures and table look fine too (I just think the table belongs in the body of manuscript).

Response: Thank you very much for your comment. The table has been incorporated in the body of the manuscript.

Reviewer 5

We thank Reviewer 5 for their thorough review of the manuscript and for the valuable feedback provided.

---

## [Editor Report · Decision Letter 2]

PONE-D-24-21114R2Emotional Skills and Health Assessment in Interventions for Intimate Partner Violence Perpetrators: A Systematic Review of Randomized Controlled TrialsPLOS ONE

Dear Dr. Aranda,

Thank you for submitting your manuscript to PLOS ONE. After careful consideration, we feel that it has merit but does not fully meet PLOS ONE’s publication criteria as it currently stands. Therefore, we invite you to submit a revised version of the manuscript that addresses the points raised during the review process.

 **Before proceed with publication, see previous reviewers required minors revisions. **

We look forward to receiving your revised manuscript.

Kind regards,

Sara Ventura

Academic Editor

PLOS ONE
---

## [Author Response · Author response to Decision Letter 3]

23 Jun 2025

Comment 1. Please review your reference list to ensure that it is complete and correct. If you have cited papers that have been retracted, please include the rationale for doing so in the manuscript text, or remove these references and replace them with relevant current references. Any changes to the reference list should be mentioned in the rebuttal letter that accompanies your revised manuscript. If you need to cite a retracted article, indicate the article’s retracted status in the References list and also include a citation and full reference for the retraction notice.

Response: Thank you for the recommendation. We have reviewed the reference list and corrected the errors identified in the numbering order of the in-text citations and corresponding references. In this third revision, aside from addressing the aforementioned issue, no citations have been added or removed. Nonetheless, we note that such changes were made during the previous revision of the manuscript (Revision 2).

A) The study by Babcock (2004) has an updated version of the meta-analysis; therefore, the citation has been replaced with the more recent one:

23. Babcock JC, Gallagher MW, Richardson A, Godfrey DA, Reeves VE, D’Souza J. Which battering interventions work? An updated Meta-analytic review of intimate partner violence treatment outcome research. Clin Psychol Rev 2024;111:102437. pmid: 38810357

B) The following new citations have been included:

31. Santirso FA, Gilchrist G, Lila M, Gracia E. Motivational strategies in interventions for intimate partner violence offenders: A systematic review and meta- analysis of randomized controlled trials. Psychosoc Interv 2020;29(3):175-190.

32. Soleymani S, Britt E, Wallace-Bell M. Motivational interviewing for enhancing engagement in intimate partner violence (IPV) treatment: A review of the literature. Aggress Violent Behav 2018;40:119-127.

33. Zarling A, Berta M, Miller A. Changes in Psychological Inflexibility and Intimate Partner Violence Among Men in an Acceptance and Commitment Therapy-Based Intervention Program. Behav Sci 2025;15:317. pmid: 40150212

53. Turner W, Morgan K, Hester M, Feder G, Cramer H. Methodological Challenges in Group-based Randomised Controlled Trials for Intimate Partner Violence Perpetrators: A Meta-summary. Psychosoc Interv 2023;32:123–136. pmid:37383642

54. Karakurt G, Koç E, Çetinsaya E. E, Ayluçtarhan Z, Bolen S. Meta-analysis and systematic review for the treatment of perpetrators of intimate partner

Comment 2. While revising your submission, please upload your figure files to the Preflight Analysis and Conversion Engine (PACE) digital diagnostic tool, https://pacev2.apexcovantage.com/. PACE helps ensure that figures meet PLOS requirements. To use PACE, you must first register as a user. Registration is free. Then, login and navigate to the UPLOAD tab, where you will find detailed instructions on how to use the tool. If you encounter any issues or have any questions when using PACE, please email PLOS at figures@plos.org. Please note that Supporting Information files do not need this step.

Response: We have proceeded to upload the Figures, as indicated in the comment, to the Preflight Analysis and Conversion Engine (PACE) digital diagnostic tool. A screenshot of the process can be seen in the attached file.

Thank you very much.

---

## [Editor Report · Decision Letter 3]

Emotional Skills and Health Assessment in Interventions for Intimate Partner Violence Perpetrators: A Systematic Review of Randomized Controlled Trials

PONE-D-24-21114R3

Dear Dr. Aranda,

We’re pleased to inform you that your manuscript has been judged scientifically suitable for publication and will be formally accepted for publication once it meets all outstanding technical requirements.

Kind regards,

Sara Ventura

Academic Editor

PLOS ONE
---

## [Editor Report · Acceptance letter]

PONE-D-24-21114R3

PLOS ONE

Dear Dr. Aranda,

I'm pleased to inform you that your manuscript has been deemed suitable for publication in PLOS ONE. Congratulations! Your manuscript is now being handed over to our production team.

Kind regards,

on behalf of

Dr. Sara Ventura

Academic Editor

PLOS ONE